# Perspectives of Next-Generation Live-Attenuated Rift Valley Fever Vaccines for Animal and Human Use

**DOI:** 10.3390/vaccines11030707

**Published:** 2023-03-21

**Authors:** Paul J. Wichgers Schreur, Brian H. Bird, Tetsuro Ikegami, Erick Bermúdez-Méndez, Jeroen Kortekaas

**Affiliations:** 1Department of Virology and Molecular Biology, Wageningen Bioveterinary Research, Wageningen University & Research, 8221 RA Lelystad, The Netherlands; 2BunyaVax B.V., 8221 RA Lelystad, The Netherlands; 3One Health Institute, School of Veterinary Medicine, University of California, Davis, CA 95616, USA; 4Department of Pathology, The University of Texas Medical Branch at Galveston, Galveston, TX 77555, USA; 5The Sealy Institute for Vaccine Sciences, The University of Texas Medical Branch at Galveston, Galveston, TX 77555, USA; 6The Center for Biodefense and Emerging Infectious Diseases, The University of Texas Medical Branch at Galveston, Galveston, TX 77555, USA; 7Laboratory of Virology, Wageningen University & Research, 6708 PB Wageningen, The Netherlands

**Keywords:** live-attenuated, vaccine, reverse genetics, Rift Valley fever virus, next-generation

## Abstract

Live-attenuated Rift Valley fever (RVF) vaccines transiently replicate in the vaccinated host, thereby effectively initiating an innate and adaptive immune response. Rift Valley fever virus (RVFV)-specific neutralizing antibodies are considered the main correlate of protection. Vaccination with classical live-attenuated RVF vaccines during gestation in livestock has been associated with fetal malformations, stillbirths, and fetal demise. Facilitated by an increased understanding of the RVFV infection and replication cycle and availability of reverse genetics systems, novel rationally-designed live-attenuated candidate RVF vaccines with improved safety profiles have been developed. Several of these experimental vaccines are currently advancing beyond the proof-of-concept phase and are being evaluated for application in both animals and humans. We here provide perspectives on some of these next-generation live-attenuated RVF vaccines and highlight the opportunities and challenges of these approaches to improve global health.

## 1. Rift Valley Fever

Rift Valley fever (RVF) is a mosquito-borne disease of ruminants, camelids and humans. High fatality ratios among young animals, mainly lambs and goat kids, and abortion storms in sheep flocks are characteristic features of RVF outbreaks. Whereas the RVF virus (RVFV) is transmitted between livestock primarily via mosquito vectors, humans can become infected either via mosquito bites or via contact with contaminated animal products, e.g., during the slaughtering of infected animals, contact with aborted fetal tissues and fluids, or the consumption of raw milk products. Most infected humans develop a self-limiting viral illness (characterized by fever, myalgia/arthralgia and general malaise). However, in a small percentage of cases, complications may develop, varying from retinal lesions to encephalitis and life-threatening hemorrhagic fever. The overall case fatality rate is estimated to range from 0.5% to 2%, but higher mortality ratios have been reported among hospitalized cases [1].

Following the first recorded outbreak of RVF in the 1930s, the virus caused large epizootics and epidemics in numerous countries across the African continent. Between 2020–2022, outbreaks have been reported in Burundi, Uganda, Niger, Libya, Mauritania, Senegal, Sudan, Kenya, and Madagascar. In most countries where RVFV emerged, the virus continues to cause epizootics and epidemics, generally separated by long inter-epizootic/epidemic periods. Stimulated by climate change and globalization, RVFV is expected to continue causing epidemics in previously unaffected regions [2], following in the footsteps of many other arthropod-borne viruses that have emerged in the past decades, including West Nile virus, Chikungunya virus, Zika virus, Usutu virus, Bluetongue virus, Schmallenberg virus and, most recently, Japanese encephalitis virus. Whereas most of these viruses cause disease in either animals or humans, RVFV is broadly pathogenic to a variety of species and the cause of serious disease in three major livestock taxa (sheep, goats, cattle), camelids, wild ruminants, rodents, and humans. Furthermore, over 40 different mosquito species have been shown in laboratory studies to be able to support RVFV replication, several of which have a global distribution [3]. 

Outbreaks of RVFV can be massive and affect millions of animals across large geographical areas. This has resulted in the classification of RVF as a disease notifiable to the World Organization for Animal Health (WOAH, formerly known as Office International des Épizooties [OIE]). RVFV is additionally classified as an “overlap” Select Agent regulated by both the US Centers for Disease Control and Prevention (CDC) and the US Department of Agriculture (USDA) and is listed as a Category A priority pathogen by the US National Institutes of Health (NIH). RVFV is also included on the World Health Organization (WHO) R&D Blueprint list of human pathogens likely to present a large-scale risk to global health in the absence of effective countermeasures. 

## 2. Current RVF Vaccine Landscape

Presently, only a few veterinary RVF vaccines are commercially available in a limited number of African countries. For humans, no fully-licensed commercial vaccine is available. The first live-attenuated RVFV vaccine developed for use in livestock was based on the Smithburn strain, a strain isolated by K. C. Smithburn after intracerebral passaging (>100 times) of the virulent Entebbe strain in mice [4]. The Entebbe strain originated from Uganda and was isolated from a pool of *Aedes* and *Eretmapodites* mosquitoes. Although vaccines based on the Smithburn strain are the most widely used RVF vaccines in Africa, they are considered unsafe for specific ruminant breeds and may cause teratogenic effects when administered during gestation [5,6,7]. 

The isolation of Clone 13, a plaque-purified clone with a naturally occurring deletion of 69% (549 nucleotides) of the NSs gene, represented another milestone in RVF vaccinology. The Clone 13 vaccine was shown to be highly efficacious in three target species (sheep, goats, cattle), including pregnant animals [8]. However, a later study in which an overdose of Clone 13 was applied in pregnant ewes showed a strong association with congenital malformations and fetal stillbirths and demonstrated that the virus could cross the placental barrier [9]. 

Inactivated vaccines for RVFV have also been commercialized and can be utilized safely in pregnant animals. However, the optimal efficacy of these vaccines depends on two initial immunization doses and yearly booster vaccinations. Significant progress has also been made with the development of vaccines based on viral vector platforms, although none have yet achieved licensure. For a more complete and extensive overview of RVF vaccine research and development efforts, we refer to the following reviews [10,11,12,13]. Here, we specifically focus on the rationale and technology supporting the development of next-generation live-attenuated RVF vaccines and provide our joint perspectives on these platform technologies.

## 3. Increased Understanding of the RVFV Life Cycle Resulted in the Construction of Safe and Efficacious Next-Generation Live-Attenuated RVFV Vaccine Candidates Using Reverse Genetics

In the past decades, our understanding of the RVFV infection and replication cycle has improved considerably. Individual viral and host factors involved in the virus life cycle have been identified, and the complex interplay between viral and host components is currently being dissected at high resolution by multiple-omics approaches. By combining the ability to artificially modify the RVFV genome through reverse genetics, already in place for >15 years, and new insights into RVFV entry and replication, various novel RVFV recombinant viruses have been constructed. Several of these recombinant viruses are highly promising candidate vaccines, as further detailed in Section 4.

### 3.1. RVFV Reverse Genetics

Classical live-attenuated RVF vaccines were generated by stochastic, laborious processes involving mutagen exposure, plaque selection procedures and/or sequential passaging in non-natural hosts. With these selection methods, there is no control over the specific mutations that occur, making it challenging to determine to what extent each individual mutation contributes to phenotypic changes and attenuation and whether these mutations could revert back to the virulent phenotype. Using reverse genetics, viruses lacking virulence factors, containing rationally selected point mutations, interchanged untranslated regions (UTRs), or modified di-nucleotides or codons can be created relatively easily. The only required feature of such variants is their ability to replicate in the cells used for rescue and manufacturing. Currently, several RVFV reverse genetics approaches are available based on either T7 polymerase or RNA Polymerase I [14,15,16]. For a comprehensive review of bunyavirus reverse genetics systems, including RVFV reverse genetics, we refer to a recent review paper [17].

### 3.2. Role of RVFV Non-Structural Proteins

The RVFV genome is divided over three RNA segments (S, M and L) of negative polarity and is about 12 kb in length. The genome encodes for the RNA-dependent RNA polymerase, a nucleocapsid (N) protein and the structural glycoproteins Gn and Gc. Additionally, the genome encodes two non-structural proteins. The non-structural protein NSs, encoded by the antigenomic S-segment, is the most important virulence factor. The main role of NSs is to antagonize the host type I interferon response, which is triggered by RIG-I-dependent recognition of the RVFV genomic panhandle structures [18]. NSs was shown to promote the post-translational degradation of the TFIIH p62 subunit, which leads to cellular transcriptional cessation, including type I IFN genes [19], and to interfere with chromosome cohesion and segregation [20]. NSs is associated with the virulence of RVFV in various species, including mice, rats and ruminants [21,22,23,24,25]. Importantly, NSs is not needed for replication in interferon-incompetent cells in vitro but is essential for the virus to cause viremia and disease in interferon-competent animals [24]. Interestingly, NSs is also not essential for replication in mosquitoes [26]. The second non-structural protein of the virus is encoded by the M-segment and is hence called NSm, which was shown to prevent apoptosis by yet unknown mechanisms [27]. Moreover, viruses lacking NSm were shown to be strongly attenuated in mice and replicated to reduced titers in murine macrophages [28]. Deletion of the NSm coding region severely restricted mosquito midgut infection and was suggested to compromise vector-borne transmission potential [26,29]. Finally, an extended Gn protein which is transcribed from the first AUG codon of the M-segment (instead of the fourth AUG codon) that is referred to as the P78 or Large Glycoprotein (LGp), was shown to be a major determinant of virus dissemination in mosquitoes [28]. Whether this protein also contributes to virulence in mammals is not fully clear, although the most recent report suggests that virulence in mice is affected by the lack of P78 expression [30].

### 3.3. RVFV Exhibits Remarkable Genome Packaging Flexibility 

The UTRs flanking each genome segment play an important role in genome packaging, as they are involved in the formation of viral ribonucleoproteins (RNPs). RNPs are believed to interact with the cytoplasmic tail of the Gn protein, facilitating genome segment incorporation into virions [31,32]. The eight-segmented influenza A virus employs a highly selective genome packaging process facilitated by the formation of a supramolecular RNP complex comprising all eight genome segments. In contrast to the influenza A virus, RVFV does not selectively incorporate all genome segments (S, M and L) into each virion. RVFV packages its genome segments randomly, resulting in a heterogeneous population of progeny virions [33,34,35]. The non-selective nature of RVFV genome packaging was first evidenced by experiments in which individual genome segments were visualized inside infected cells and inside virions using fluorescent probe sets able to discriminate between the segments. This so-called single-molecule fluorescence in situ hybridization technique (smFISH) revealed the absence of the formation of a RVFV supramolecular RNP complex and provided data showing that only a minor fraction of virions contain all three genome segments. The majority of virions lack either one, two, or even all three genome segments, rendering them incapable of spreading autonomously [33,35]. Interestingly, any theoretical fitness cost apparently associated with an inefficient genome packaging process may, at least partially, be compensated by the co-infection of cells with multiple incomplete particles. Supporting this hypothesis, a recent study showed that two distinct virus particle populations with incomplete genomes efficiently complemented each other upon co-infection, resulting in the rescue of infectious viruses. This suggests that incomplete particles can contribute to viral spread and should consequently not be considered defective interfering particles [36].

### 3.4. Exploiting Flexibility in RVFV Genome Organization and Packaging for Vaccine Development

The flexibility in RVFV genome packaging, the limited role of the non-structural proteins in in vitro growth in cells with compromised innate immune signaling, and the development of robust reverse genetics systems, as detailed above, have facilitated the generation of a wide range of genetically engineered RVF viruses with attenuated or even avirulent phenotypes (Figure 1). Variants were created in which the NSs gene is deleted or replaced for a gene encoding a fluorescent marker protein such as eGFP. These variants are still able to grow efficiently in IFN-deficient cells [37] but were shown to be strongly attenuated in vivo [38]. Moreover, variants were created that lacked NSm and/or LGp expression with or without an NSs deletion [22]. Furthermore, an autonomously replicating two-segmented RVFV was successfully rescued as well. To generate this variant, the glycoprotein precursor gene was expressed from the NSs locus of the S-segment, resulting in a virus lacking an M-segment [39]. Notably, variants with a completely reconfigured S-segment, in which eGFP was expressed from the N locus and the N protein from the NSs locus, were also shown to be viable [40]. Finally, one of the latest examples showing a high degree of RVFV genome plasticity and genome packaging flexibility was the successful rescue of four-segmented RVFV (RVFV-4s) [41]. RVFV-4s contains an authentic L RNA segment, two M-type RNA segments, one encoding Gn and one encoding Gc, and either a full-length or NSs deleted S-segment.

## 4. Current Status of Next-Generation Live-Attenuated RVF Vaccine Development

Several next-generation live-attenuated candidate vaccines have been developed utilizing the technological approaches described and are currently being further assessed for veterinary and/or human use. 

### 4.1. MP-12 and MP-12 Deletion Variants

Besides the Smithburn and Clone 13-based vaccines, which are commercially available for livestock, the live-attenuated MP-12 vaccine is probably one of the best-characterized RVF vaccines to date. The efficacy of MP-12-based vaccines is outstanding with a single dose, similar to the Clone 13 and Smithburn vaccines; however, safety concerns have also been noted for this vaccine, including fetal malformation in ewes vaccinated during the first trimester [42] and mild transient hepatocellular necrosis in vaccinated lambs and calves [43,44]. Though MP-12 is not yet commercially available, the vaccine is currently conditionally licensed for ruminant livestock in the United States. MP-12 has also been evaluated in phase 1 and 2 clinical trials in human volunteers [45,46]. 

MP-12 was developed by the serial passage (#12) and plaque cloning of the virulent wild-type ZH548 strain in human diploid lung MRC-5 cells in the presence of the chemical mutagen 5-fluorouracil. MP-12 was found to contain four, nine, and 10 mutations in the S, M, and L segments, respectively. At least three amino acid substitutions in the M and L segments (Gn-Y259H, Gc-R1182G, and L-R1029K) were shown to contribute to the attenuation of the strain [47]. Combined reversed mutations at those three sites (Gn-H259Y, Gc-G1182R, and L-K1029R) partially recovered virulence in a mouse model, though 80% of inoculated animals survived, indicating that the attenuation phenotype of the MP-12 strain is obtained by a combination of multiple mutations, including those three sites. Notably, MP-12 replication is restricted at 38 °C and above due to four temperature-sensitive mutations in the M and L segments (Gn-Y259H, Gc-R1182G, L-V172A, and L-M1244I) [48]. Among them, the L-M1244I mutation was not stable following 25 serial viral passages at 37 °C in Vero or MRC-5 cells [49]. Consequently, the genetic integrity of the MP-12 vaccine should be ensured by maintaining an appropriate seed lot system and culture temperature. 

For all RVFV live-attenuated vaccine candidates, including MP-12, mosquito-borne transmission has been cited as a potential concern [50]. However, although the MP-12 strain retains the ability to infect and disseminate in mosquitoes via the oral intake of blood meals, transient low-level viremia (~10^3^ PFU/mL) in vaccinated animals is not considered sufficient for viral transmission to and dissemination in mosquitoes [50,51]. Furthermore, if reassortant strains emerge, which is so far only a theoretic possibility, through the combination of MP-12 and a circulating wild-type RVFV in animals or mosquitoes, the reassortant would have a virological phenotype like that of the circulating wild-type RVFV and would not create an additional safety risk [52]. A comprehensive overview of the safety and efficacy of MP-12-based vaccine candidates can be found in the following review paper [53]. 

To improve the safety profile of MP-12, several research groups constructed MP-12 variants lacking one or more of the virulence genes using reverse genetics or even reconfigured the genome to a two-segmented variant [15,27,54]. Two MP-12 variants referred to as arMP-12ΔNSs16/198 and arMP-12ΔNSm21/384, which lack an intact NSs and P78/NSm gene, respectively, could induce protective immunity in 3 to 4-month-old young sheep, 4 to 6-month-old cattle, or pregnant ewes without safety concerns [55,56,57]. Notably, the neutralizing antibody responses following inoculation with arMP-12ΔNSm21/384 were stronger compared to those induced by arMP-12ΔNSs16/198 [58]. A large-scale follow-up experiment with 17 pregnant African domestic sheep vaccinated with arMP-12ΔNSm21/384 during the early stage of pregnancy (<35 days gestation) resulted in fetal malformations (forelimb malformation and deformed tail) in 18% of vaccinated ewes [58]. At post-mortem, the tissues of dead lambs (spleen, lung, brain, and long bone) were negative for RVFV as analyzed by PCR. Nevertheless, these results indicated that the arMP-12∆NSm21/384 vaccine should not be used in pregnant sheep during the first month of gestation. 

More recently, the next-generation MP-12 strain, RVax-1, was generated and characterized [59]. The RVax-1 candidate vaccine encodes 36, 37, 167, and 326 silent mutations in the N, NSs, M, and L ORFs, respectively, in the backbone of arMP-12ΔNSm21/384. A cluster of silent mutations was introduced at every 50 nucleotides within the ORFs, while the codon usage or codon pair bias was not deoptimized to maintain efficient viral translation. All 566 silent mutations were genetically stable across 10 serial passages in Vero cells, and especially the silent mutations in the S- and L-segment were shown to strengthen the attenuated phenotype, compared to MP-12 [60]. Following direct recovery from Vero cells using polymerase I-based reverse genetics [14], RVax-1 was shown to replicate efficiently in Vero and MRC-5 cells and to have comparable immunogenicity and protective efficacy compared to the full-length rMP-12 strain in a mouse model. Moreover, the RVax-1 strain, but not the full-length rMP-12, was shown to poorly disseminate in orally fed *Aedes aegypti* mosquitoes, indicating a minimum risk of mosquito-borne transmission of RVax-1 [59]. The silent mutations unique to RVax-1 could furthermore serve as a genetic marker to trace vaccine RNA in mosquitoes and livestock animals. Further evaluations of vaccine safety, immunogenicity and efficacy will support the development of this next-generation MP-12 candidate vaccine.

### 4.2. DDvax

Stimulated by the 2006–2007 East Africa RVF epidemic [61], an effort to improve upon the safety and efficacy of conventional RVFV vaccines using reverse genetics systems was undertaken with the goal of developing the first rationally designed RVF vaccine candidate. Using a precise whole gene deletion strategy, the two main RVFV virulence factor genes (the NSs and NSm coding regions) were removed from the parental ZH-501 strain. These deletions resulted in a highly attenuated and highly immunogenic vaccine virus particle initially designated as DNSs-DNSm-rZH501, and is currently referred to simply as “DDvax” for “double-deletion vaccine” [22]. 

Initial testing in rats revealed a high safety profile with no obvious clinical adverse events after inoculation of DDvax at doses exceeding 100,000× the known lethal dose 50 (LD_50_) of the parental RVFV strain. Complete protection from high-dose virulent virus challenge was observed 28 days post-vaccination. Over the past 16 years since that initial study, DDvax has been proven safe, immunogenic, and effective in preventing virulent virus infection and disease following a single dose administration in a variety of animal species, including multiple rodent species, adult pregnant and non-pregnant livestock species, and two species of nonhuman primates (marmosets and rhesus macaques) [23,62,63] (and manuscripts in preparation). In each animal species tested, robust and rapid rises in neutralizing antibodies were observed from 14 to 21 days post-vaccination. Surprisingly, results in rodents have indicated that rapid protection from virulent virus infection is conferred in as little as 2–3 days post-vaccination (manuscript in preparation). This early protection was observed before the onset of detectable neutralizing antibodies and may be conferred by the stimulation of robust innate cellular responses (e.g., interferon and other mediators) linked to de novo vaccine replication and protein synthesis. 

Further preclinical and research and development activities with the underlying DDvax technological platform have also revealed insights into RVFV mosquito infection and transmission factors using various single (NSs or NSm only) and double (NSs and NSm) whole gene deletion recombinant viruses. As part of the safety assessments of DDvax and these other gene deletion RVF viruses, initial mosquito transmission experiments revealed that the NSm gene (and/or P78) confers critical mosquito mid-gut barrier evasion mechanisms that, when deleted, provide an additional environmental containment safety factor. Across several studies, the DDvax research team has definitively demonstrated that DDvax cannot be efficiently transmitted by multiple mosquito vectors due to the specific deletion of the NSm gene region [26,29,64]. These results, when taken together, indicated that DDvax might provide in a single dose rapid, safe, and robust protection from RVFV infection with an excellent environmental safety profile that would potentially enable use in both routine and emergency outbreak response-related vaccination campaigns. 

### 4.3. RVFV-4s

RVFV-4s comprises four instead of the natural three-genome segments, resulting from splitting the M-segment into an M-type segment encoding NSm and Gn and an M-type segment encoding Gc. Several RVFV-4s variants have been constructed, including variants encoding eGFP [41], though for vaccine purposes, two RVFV-4s strains have been brought beyond the proof-of-concept phase: vRVFV-4s for veterinary use and hRVFV-4s for human use. vRVFV-4s is based on wild-type strain 35/74 isolated from a sheep, whereas hRVFV-4s is based on Clone 13, which was derived from wild-type strain 74HB59, isolated from a human case. Both vaccine candidates lack an intact NSs gene, in addition to a split M-segment. The vaccine viruses were shown to be genetically stable (>20 passages), to not revert to virulence and to not disseminate within a mammalian host or spread to the environment [65]. 

The attenuated phenotype of RVFV-4s, comprising four genome segments instead of three, is explained by the lower chance of incorporating a complete set of genome segments and an imbalance in replication and transcription [41]. A single vaccination with vRVFV-4s was shown to induce robust levels of neutralizing antibodies within 1–2 weeks after vaccination in all species evaluated (sheep, goats, cattle) [65]. The initial safety of the hRVFV-4s vaccine was assessed in mice following intranasal administration. In contrast to Clone 13, which can induce lethal encephalitis when administered intranasally in mice, hRVFV-4s was shown to be completely avirulent and unable to cause viremia [66]. High-dose experiments with young lambs (intramuscular route), which are the most susceptible target animals of RVFV, subsequently demonstrated an absence of viremia, absence of shedding and spreading and confirmed that RVFV-4s could not be passaged from animal to animal and does not revert to virulence [65]. Furthermore, a recent experiment with marmosets, a nonhuman primate species highly susceptible to the wild-type virus, showed that vRVFV-4s and hRVFV-4s do not induce untoward effects nor disseminate beyond draining lymph nodes [67]. Finally, RVFV-4s was shown to be safe for pregnant ewes, with the absence of vertical transmission during the first trimester of gestation [68]. Additional studies assessing (reproductive) toxicology and potential virus dissemination in mosquitoes are pending. 

## 5. Beyond Proof-of-Concept Challenges 

Following the initial assessment of the candidate vaccines for safety and efficacy in the available animal models, the route to vaccine registration is a long and rigorous process. Not only industrial scale production pipelines should be in place, but multiple studies confirming stability, potency, adventitious agent exclusions, and other safety assessments should be conducted, where appropriate, under good laboratory and manufacturing practices (GLP, GMP) to provide a clear path towards regulatory approvals. 

### 5.1. Production of the Live-Attenuated RVF Vaccine Candidates

In the industrial scale development process, various parameters such as time of infection, time of harvest and multiplicity of infection should be optimized, and the absence of adventitious agents should be confirmed, with serum-free processes preferred over those containing products of animal origin. For adventitious agent testing, validated next-generation sequencing (NGS) pipelines could be the future, though, at this moment, they are not yet broadly available. Nevertheless, the use of NGS-based adventitious agent testing to replace conventional assays is encouraged by the European Pharmacopeia (relevant chapters 5.2.14, 2.6.16), the US Food and Drug Administration (FDA) [69] and the WHO [70,71,72]. Finally, vaccine stability should be optimized to prolong shelf-life and minimize cold-chain requirements. In contrast to more targeted approaches (e.g., mRNA platforms), multiple antigens are being presented to the immune system at once, including Gn, Gc, N and the polymerase, thereby facilitating a broader immune response to a variety of immunologically relevant antigens. Furthermore, the production of live-attenuated vaccines can be cost-effective as, typically, no adjuvants or lipid nanoparticles are required. 

### 5.2. Assessing Vaccine Potency and Safety 

To be able to assess and monitor vaccine efficacy, it is very important to have appropriate diagnostic tests in place that can measure potential breakthrough infections as well as a correlate of protection. The primary recognized correlate of protection for RVFV are detectable levels of neutralizing antibodies. The main target of neutralizing antibodies are the glycoproteins on the outer envelope of the virus. The most effective are those antibodies directed towards the head domain of the Gn protein [73,74,75,76]. For RVFV, the virus neutralization test based on a virulent virus is considered a gold standard for the detection of neutralizing antibodies; however, progress is being made with neutralization tests using attenuated strains, virus-like particles or pseudotyped virus particles that can be performed under BSL-2 conditions [77]. In humans, clear T-cell-based correlates of protection have yet to be described, though a recent study indicated a clear role for cellular immunity in protection through the interaction of various cellular components [78]. The concept for any RVF vaccine to differentiate naturally infected from vaccinated individuals or animals (e.g., DIVA) is a diagnostic challenge that may best be assessed as a part of large-scale clinical trials. In animals, a focus at the herd- or flock levels may be most appropriate in endemic areas, as antibodies against non-structural proteins, currently the only putative DIVA targets, are generally short-lived [79,80]. 

A final challenge in the registration of RVFV vaccines is the demonstration of safety for the most susceptible target species at the most susceptible physiological stage. Pregnant animals and newborns are the most susceptible to severe disease, which therefore have to be included in such studies. Specifically, safety for pregnant animals should be assessed at each trimester of gestation, and pregnancy should be brought to full term. While several newly-developed live-attenuated RVF vaccines have shown promise in limited studies with young and pregnant ruminants, performing all required studies with quality standards required for vaccine registration is a costly endeavor. 

## 6. Target Product Profiles and Pathways toward Vaccine Licensing and Availability

With the availability of candidate vaccines that were shown efficacious and safe for even the most susceptible target species, a next-generation RVF veterinary vaccine and the first human vaccine could be brought to market in the coming years. To facilitate this process, and as part of the WHO R&D Blueprint global strategy, Target Product Profiles (TPPs) have been drafted, providing early technical guidance for the development of these vaccines. These draft TPPs describe vaccine characteristics for “optimal targets” and “minimally acceptable targets”, including indications of use, target populations, safety and efficacy requirements, vaccine stability and shelf life. For RVF vaccines, three separate TPPs were proposed by the WHO: one for human vaccines for emergency use, one for vaccines for the protection of persons at high risk of RVFV infection, and one for animal vaccines [81]. 

### 6.1. Human Vaccines

The WHO TPPs for human vaccines state that vaccines should be WHO-prequalified [82]. By obtaining WHO prequalification (PQ), the safety and efficacy of vaccines is guaranteed through regular re-evaluation, production site inspections, and investigation of adverse events following immunization. Before a vaccine can be WHO-prequalified, the manufacturers need to have obtained marketing authorization from the national regulatory authority of the country of manufacture. Despite the availability of promising candidate vaccines and the expectancy of new epidemics, pharmaceutical companies have historically been reluctant to invest in human RVF vaccines. To overcome this, the Coalition for Epidemic Preparedness Innovations (CEPI) is supporting the clinical development of several RVF vaccine candidates. However, with the low incidence of endemic RVF and epidemics occurring sporadically, a major challenge that remains is the organization of phase 3 studies for the assessment of vaccine performance under field conditions. Considering that this challenge is relevant not only to RVF but to all diseases on the WHO Blueprint list, an innovative solution is needed in the regulatory field to allow marketing authorization when conventional phase 3 studies are not feasible. 

Streamlined with the WHO PQ program, the European Medicines Agency (EMA) has developed the EU-M4all procedure (previously known as Article 58) to promote the development of vaccines for people in low- and middle-income countries outside the EU. This procedure facilitates collaborations of the Committee for Medicinal Products for Human Use (CHMP) with the disease expertise of WHO and national regulators of the target countries to facilitate the prequalification of the vaccines by WHO and registration in target countries. 

A potential path towards licensing and availability of a human RVF vaccine was paved by the Ebola vaccine that was developed by Merck Sharp & Dohme with financial support from the US government. After accelerated development in response to the 2014–2016 Ebola outbreak in West Africa and subsequent “compassionate use”, the vaccine was prequalified by WHO and is now licensed in Europe, the US and eight African countries. Subsequently, a stockpile of the vaccine has been made available through the International Coordinating Group on Vaccine Provision with financial support from Gavi, the Vaccine Alliance. A similar overall strategy might be feasible for a human RVF vaccine, thereby optimally preparing for a future outbreak. 

### 6.2. Veterinary Vaccines

The WHO-recommended TPP for veterinary RVF vaccines includes sheep, goats, cattle, and dromedary camels as target populations. Demonstrating safety and efficacy in all these different species makes the path toward licensure challenging and may impact commercial interest for further development, given the uncertain marketplace for RVF animal vaccines. In Europe specifically, to improve the availability of veterinary medicines for the treatment of minor animal species and the prevention of uncommon diseases of major animal species, the Committee for Medicinal Products for Veterinary Use (CVMP) of EMA has developed a guideline for veterinary medicinal products. These include the treatment or prevention of diseases that occur infrequently, occur in limited geographical areas, or affect animal species other than cattle, sheep, pigs, chickens, dogs and cats, previously referred to as the minor use/minor species (MUMS) policy. This policy provides two incentives to develop products that would otherwise not be developed in the current market conditions: manufacturers of vaccines that are considered within the scope of this regulation benefit from reduced data requirements for marketing authorization regarding safety, efficacy, and quality. The second incentive is financial, by means of fee reductions or exemptions. The regulation is known as the “Limited Market” Regulation (Article 23 of Regulation 2019/6), coming into effect on 28 January 2022 [83]. Specific to European registration of next-generation RVF vaccines, these types of vaccine candidates may meet the requirements to be classified as Limited Market in line with Articles 4 (29) and 23 of Regulation 2019/6. If so, demonstrating efficacy through laboratory (pre-clinical) efficacy studies and omitting large-scale field trials may be sufficient as long as required data from such studies is provided. In addition, GLP requirements of pre-clinical safety studies (i.e., safety in pregnant animals) could be lifted if protocols and reports allow a satisfactory assessment of the trials. Although further requirements for safety within the Limited Market procedure may be reduced, considering that the vaccines described here are genetically modified organisms, the requirements of Directive 2001/18/EC (on the deliberate release into the environment of genetically modified organisms) must be met. It should be noted that these regulations and recommendations would not necessarily apply to any registrations of vaccines outside of the EU regulatory framework.

Despite the availability of next-generation candidate vaccines with outstanding safety and efficacy profiles and advances in the veterinary regulatory landscape that could reduce the costs of vaccine licensing significantly, a main barrier to comprehensive vaccination strategies is the lack of strong market incentives due to the sporadic and unpredictable nature of RVF epizootics. To help overcome these challenges, financial support mechanisms driven by the international veterinary community could be employed and may prove essential to bring next-generation veterinary RVF vaccines to market. 

## 7. Conclusions

The knowledge gained in the past 20 years on the molecular biology of RVFV and the availability of reverse genetics systems has resulted in the development of several highly promising live-attenuated candidate vaccines potentially suitable for both human and animal use. However, the steps towards full licensure and the eventual inclusion of RVFV vaccines into routine or emergency vaccination programs will require collective effort and support from international health organizations, regulatory authorities, and clearly defined end-users. This work is critical to ensure that next-generation RVF vaccines are available and ready for use before the next RVF epidemic. RVF is a classic One Health zoonotic pathogen, and the impact of the disease touches on all aspects of human and animal health and food security. The time is now to enhance our global capacities to mitigate this threat to health security.

## Figures and Tables

**Figure 1 vaccines-11-00707-f001:**
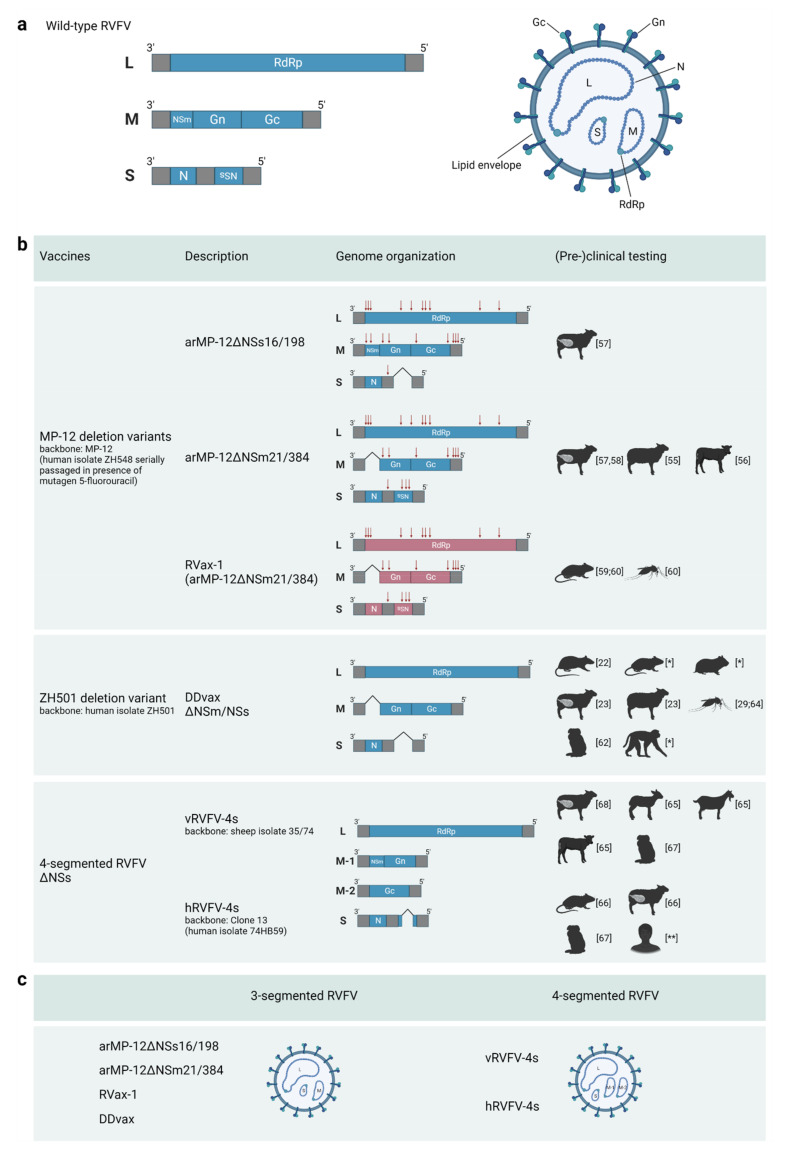
Schematic representation of the genome organization of wild-type RVFV and live-attenuated vaccine candidates currently being developed beyond the proof-of-concept phase. (**a**) Wild-type RVFV genome and virion. (**b**) Genome organization of a selection of live-attenuated RVF vaccine candidates. Red arrows indicate the location of point mutations in a particular genome segment, and red open reading frames in RVax-1 indicate the presence of silent mutations every 50 nucleotides. Species in which the safety and/or efficacy of the candidate vaccines have been evaluated are illustratively represented. * Manuscript in preparation, ** Ongoing trial. (**c**) Classification of live-attenuated RVF vaccine candidates as 3- or 4-segmented viruses.

## Data Availability

No new data were created or analyzed in this study. Data sharing is not applicable to this article.

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
