# Peer review of "Perspectives of Next-Generation Live-Attenuated Rift Valley Fever Vaccines for Animal and Human Use"

_vaccines, 2023, doi:10.3390/vaccines11030707_

Round 1

Reviewer 1 Report

This review by Wichgers Schreur et al. describes the generation, efficacies and safety profiles of several live-attenuated RVF vaccines. It also discusses the vaccine production process, licensing and financing of such vaccines for human and veterinary use. 

Given that the authors have a long-standing expertise in the development and evaluation of RVF vaccines, the manuscript is obviously written to a very high standard. Although I was familiar with most of the concepts already, I believe that it is accessible and interesting to the non-expert reader. The "Beyond proof-of-concept challenges" section was particularly insightful and perhaps adds some novelty to the review. 

I recommend publication without revision. I have only a few very minor comments that the authors may wish to consider:

- Lines 129/130 and 174/175: please add references

- Lines 242-244/245: I have had to read this paragraph multiple times but was still not sure how RVax-1 differs from rMP12-GM50. Further, the paragraph makes reference to "silent mutations (at) every 50 nucleotides" and it is unclear where this comes from. I assumed this is how the prototype strain is constructed based on its name but I had to look up the reference to be sure. Could this be simplified?

- Line 256: What do you refer to by "lessons learned during the 2006/2007 East Africa epidemic"?

- Line 317: "does not" should read "do not"

- Line 320: "neither" should read "or"

- Line 366: delete "also"

Author Response

Reviewer #1 (Comments to the Author):

This review by Wichgers Schreur et al. describes the generation, efficacies and safety profiles of several live-attenuated RVF vaccines. It also discusses the vaccine production process, licensing and financing of such vaccines for human and veterinary use. 

Given that the authors have a long-standing expertise in the development and evaluation of RVF vaccines, the manuscript is obviously written to a very high standard. Although I was familiar with most of the concepts already, I believe that it is accessible and interesting to the non-expert reader. The "Beyond proof-of-concept challenges" section was particularly insightful and perhaps adds some novelty to the review. 

I recommend publication without revision. I have only a few very minor comments that the authors may wish to consider:

- Lines 129/130 and 174/175: please add references

Response: In the revised version of the manuscript we added relevant references to these lines.

- Lines 242-244/245: I have had to read this paragraph multiple times but was still not sure how RVax-1 differs from rMP12-GM50. Further, the paragraph makes reference to "silent mutations (at) every 50 nucleotides" and it is unclear where this comes from. I assumed this is how the prototype strain is constructed based on its name but I had to look up the reference to be sure. Could this be simplified?

Response: We appreciate the comment of the reviewer and simplified part of this section for extra clarification.

- Line 256: What do you refer to by "lessons learned during the 2006/2007 East Africa epidemic"?

Response: We refer to the following paper Lessons from the 2006–2007 Rift Valley fever outbreak in East Africa: implications for prevention of emerging infectious diseases | Future Virology (futuremedicine.com) with it main message: “capacity building in both the human and animal health sectors will improve our preparedness and prevention strategies for human illness associated with impending zoonotic outbreaks”

In the revised manuscript we added the reference and reformulated our statement.

- Line 317: "does not" should read "do not"

Response: In the revised version of the manuscript we now refer to RVFV-4s instead of vRVFV-4s and hRVFV-4s.

- Line 320: "neither" should read "or"

Response: Changed as suggested

- Line 366: delete "also"

Response: Changed as suggested

Reviewer 2 Report

In this manuscript, authors review the next-generation attenuated RFV vaccine focusing on the rationale and technology. Authors commented on the history and current status of RFV vaccine development, and  that the genetically modified vRVFV-4s and/or hRVFV-4s seems to be the potential ultimate solution.

During the development of vaccine for RVFV, a common problem  on target animals, in particular ewe/lamb are the malformations, stillbirths and abortion, when applied during the first trimester of gestation. This is difficult to avoid because in mammals, the development of a placenta barrier (6 layers of cells for syndesmochorial ruminants) is usually incomplete until the end of first trimester (refer to Geisert RD and Spencer TE, placenta in mammals, advances in anatomy, embryology and cell biology).  Here also, several experimental animals, mice, rats, primates and ruminants are used, authors should also consider the differences in the placenta structure of these different species used on characterizing the pathogenicity of generated vaccine candidates. Ideal results gain on one experimental animal species do not justify ideal results on target animals, and  should be interpreted with caution.

Lines 315-316, please confirm that whether the experiment on lambs are via intranasal.  Please also indicate the results when tested on pregnant ewes during the first trimester. 

Tens of strains and mutants are mentioned in the manuscript, for a glance view, I strongly suggest authors create a big table (to accompany Figure 1) to list the strain name, manipulation (correlate with the order listed in Figure 1), most importantly the key outcome in tested animals, and the tested animal species.  Figure 1 is good, but it only focus on the genetic configuration, without the tested results.

Author Response

Reviewer #2 (Comments to the Author):

In this manuscript, authors review the next-generation attenuated RFV vaccine focusing on the rationale and technology. Authors commented on the history and current status of RFV vaccine development, and  that the genetically modified vRVFV-4s and/or hRVFV-4s seems to be the potential ultimate solution.

During the development of vaccine for RVFV, a common problem  on target animals, in particular ewe/lamb are the malformations, stillbirths and abortion, when applied during the first trimester of gestation. This is difficult to avoid because in mammals, the development of a placenta barrier (6 layers of cells for syndesmochorial ruminants) is usually incomplete until the end of first trimester (refer to Geisert RD and Spencer TE, placenta in mammals, advances in anatomy, embryology and cell biology).  Here also, several experimental animals, mice, rats, primates and ruminants are used, authors should also consider the differences in the placenta structure of these different species used on characterizing the pathogenicity of generated vaccine candidates. Ideal results gain on one experimental animal species do not justify ideal results on target animals, and  should be interpreted with caution.

Response: We appreciate the comment of the reviewer and agree that additional discussion about safety of these type of vaccines, especially during pregnancy and in different animal species should be discussed. In the revised version of the manuscript we added a short paragraph to the “Assessing vaccine potency and safety”  section.

Lines 315-316, please confirm that whether the experiment on lambs are via intranasal.  Please also indicate the results when tested on pregnant ewes during the first trimester.

Response: In both experiments the animals were intramuscularly vaccinated and intravenously challenged. In the revised version of the manuscript we added this extra information.

Tens of strains and mutants are mentioned in the manuscript, for a glance view, I strongly suggest authors create a big table (to accompany Figure 1) to list the strain name, manipulation (correlate with the order listed in Figure 1), most importantly the key outcome in tested animals, and the tested animal species.  Figure 1 is good, but it only focus on the genetic configuration, without the tested results.

Response: We appreciate the point of the reviewer and decided to add an extra column to Figure 1B to emphasize in which species a specific vaccine has been tested so far.